# The Evolution of the 3D-Printed Drug Delivery Systems: A Review

**DOI:** 10.3390/pharmaceutics14071312

**Published:** 2022-06-21

**Authors:** Ildikó Bácskay, Zoltán Ujhelyi, Pálma Fehér, Petra Arany

**Affiliations:** 1Healthcare Industry Institute, University of Debrecen, Nagyerdei körút 98, H-4032 Debrecen, Hungary; arany.petra@euipar.unideb.hu; 2Department of Pharmaceutical Technology, Faculty of Pharmacy, University of Debrecen, Nagyerdei körút 98, H-4032 Debrecen, Hungary; ujhelyi.zoltan@pharm.unideb.hu (Z.U.); feher.palma@pharm.unideb.hu (P.F.)

**Keywords:** 3D printing, drug delivery systems, tablet, implant, TTS, microneedle

## Abstract

Since the appearance of the 3D printing in the 1980s it has revolutionized many research fields including the pharmaceutical industry. The main goal is to manufacture complex, personalized products in a low-cost manufacturing process on-demand. In the last few decades, 3D printing has attracted the attention of numerous research groups for the manufacturing of different drug delivery systems. Since the 2015 approval of the first 3D-printed drug product, the number of publications has multiplied. In our review, we focused on summarizing the evolution of the produced drug delivery systems in the last 20 years and especially in the last 5 years. The drug delivery systems are sub-grouped into tablets, capsules, orodispersible films, implants, transdermal delivery systems, microneedles, vaginal drug delivery systems, and micro- and nanoscale dosage forms. Our classification may provide guidance for researchers to more easily examine the publications and to find further research directions.

## 1. Introduction

Three-dimensional (3D) printing was developed more than 30 years ago to manufacture 3D objects based on a digital design. This layer-by-layer process enables a fast and cheap design cycle for the preparation of personalized medication [1]. The term 3D printing was coined as an umbrella term and encompasses a number of processes, and in many reviews the main types were described in detail [2,3,4,5]. Three-dimensional printing gave the means to the manufacture of a high-quality product within minutes in an easy manufacturing cycle. This on-demand manufacturing was time and material saving. Not to mention the fact that 3D printers could conquer the traditional manufacturing regime of ‘one size fits all’ [6]. As 3D printing was based on a computer-aided design (CAD), it provided the ability to quickly create and produce a flexible and innovative product [7]. Personalized medication carried the opportunity to create drug delivery systems for patient’s requirements. Furthermore, 3D printing gained access to the creation of unique dosage forms and achieving more complex drug release profiles [8]. The image could be made to meet the patient’s individual needs regarding their age, weight, organ function, and severity of disease [4]. The application of 3D printing technology might be an alternative way to construct effective, customized active pharmaceutical ingredient (API) combinations for the patient immediately [9]. The 3D printing technique opened up the opportunity for the development of tailored single and multi-drug products at the point-of-care [10].

In recent years, many comprehensive publications have been presented on the different designed drug dosage forms. As Moulton et al. highlighted, this kind of process created the opportunity for the manufacturing of controlled and modified release of the APIs, enabled the delivery of poorly water soluble drugs, increased drug stability, and reduced the used API amount without compromising the efficacy [11]. In 2018, two distinct research groups summarized the recent achievements in the manufacturing of pharmaceuticals but as a rapidly developing area the achievements vary from year to year [12,13]. Mohapatra et al. gathered together the newest publications in recent years and grouped the research based on the type of 3D printing [14].

The available reviews mostly focus on one or multiple drug dosage forms manufactured by one type of 3D printing technology. Cunha-Filho et al. discussed the fabricated drug delivery systems by fused deposition modeling (FDM) 3D printing [15]. While Gueche et al. summed up the oral dosage forms created by selective laser sintering (SLS) [16] and Wang et al. described the stereolithographic (SLA) constructed oral dosage forms [17]. Inkjet printing of pharmaceuticals was summarized by Dali et al. [18]. In more and more research, three-dimensional bioprinting was used which is a new era of 3D printing technologies where researchers aim to build living tissue models [19].

During the last decades multiple research groups were established to fabricate drug delivery systems. FabRx Ltd. is one of the most innovative start-up companies, which is a biotech company, designed to produce 3D-printed medication [20]. Regarding the diverse drug dosage forms, different reviews summarized the achievements. For example buccal patches were analyzed in the work of Shirvan et al. [21], implants in the work of Domsta et al. [22], oral dosage forms in the work of Khatri et al. [23], and transdermal delivery systems (TTS) in the work of Economidou et al. [24].

The aim of this work was to provide a comprehensive image on how the manufacturing of the different drug delivery systems started and where the experiments are headed now. The chosen drug delivery systems were divided into subgroups based on the type of the drug delivery system and the tables summarized the most important research in the last 20 years in chronological order.

## 2. The 3D Printing of Drug Delivery Systems

### 2.1. Tablets

The first publication of a 3D-printed tablet dates back to 1996 when solid samples were created with a desktop printer from PCL and PEO polymers containing blue and yellow dyes. Based on the results, complex drug delivery regimes could be created with this technique, such as the release of multiple drugs or multiphasic release of a single drug. This study demonstrated several simple examples of such devices and several construction methods that could be used to control the release of the drugs [25].

In the early 3D printing articles, droplet binding was used for the manufacturing of the samples when the used binder was not necessarily polymer but some other auxiliary material, e.g., Eudragit^®^ or mannitol. The authors concluded that with this method adequate oral dosage forms can be manufactured which exhibit erosion or diffusion release mechanisms [26,27]. At the beginning of the research, the most important question for the authors was the type of the chosen additive manufacturing process, the used printing parameters, immediate- or delayed release tablet manufacturing, and first- or zero-order kinetic profile manufacturing [28].

Gbureck et al. used a unique technology for the manufacturing of the drug delivery system. Firstly, they created the sample with a 3D bioceramic powder printing process and then the used antibiotics were adsorbed during a week to fabricate the tablets [29]. Yu et al. produced an acetaminophen containing matrix tablet using a desktop 3D printer. The middle drug-containing regions of the tablets were formed by depositing the binder liquid containing release-modulation materials onto the automatically spread powder layers (Figure 1) [30]. Two years later, the same investigators decided to construct a drug delivery system from the same API, polymer, and printing technique but in this case the layers were not designed horizontally, yet vertically, to provide a different dissolution mechanism [31].

Guaifenesin-containing controlled release bilayer tablets were constructed with extrusion-based desktop 3D printing. The samples were formulated to demonstrate the production of relatively complex formulations that could mimic the release profile of a commercially available tablet [32].

Even though the wide range experiments started in the 2010s, the Food and Drug Administration agency (FDA) granted the approval of Spritam^®^ in 2015. This 3D-printed tablet was used for the treatment of epileptic seizures. This was the first and still only approved 3D-printed drug delivery system. For the manufacturing, a so-called ZipDose Technology was used [33].

Goyanes et al. published four articles in 2015 about the manufacturing of tablets for distinct purposes with the use of FDM 3D printing and PVA filament. In one of the articles, the commercially produced PVA filaments were loaded with 5- and 4-amino salicylic acid in an ethanolic drug solution. A final drug loading of 0.06% *w*/*w* and 0.25% *w*/*w* was achieved for the APIs, respectively. This filament was created with unsimilar infill percentages in a nonidentical pattern (Figure 2). The dissolution tests showed that the release profiles depended on the used infill percentage and the used API itself [34].

A polypill was designed by a RegenHU extrusion-based 3D printer which contained distinct fillable ink cartridges for the production of semi-solid API-containing materials. In this research, three different API—nifedipine, captopril, and glipizide—containing inks were manufactured by HPMC. The three APIs could be found in three diverse compartments. From the nifedipine and the glipizide-containing formulations, the drug was released by diffusion and from the captopril formulation by osmosis. The schematic image of the samples is shown in Figure 3 [36].

In the same year, a newer design was applied which is also a polypill but contained five different APIs: amino salicylic acid (ASA) and hydrochlorothiazide (HCT) in the bottom layer, and atenolol, ramipril, and pravastatin in the middle area in three separated regions. The manufacturing was described in the previous article. The ASA and HCT formulation were an immediate release compartment, while the others were controlled with a cellulose acetate membrane to provide extended release. The graphical scheme is shown in Figure 4 [37].

Metolose^®^ (a special cellulose ester) and PLA disks were created and co-extruded with nitrofurantoin as a model API. The research showed that the rheological properties depended on the amount of the undissolved particles and, as in the case of modified release tablets, the amount of the cellulose derivative affected the dissolution time from the fabricated filament [38].

In the work of Okwuosa et al., a 10% API-containing filament was produced by hot-melt extrusion. As an API dipyridamole or theophylline, as a polymer PVP, and as an excipient plasticizer was used. These constructed filaments were then FDM 3D printed. The novelty of this work is that, for the extrusion, thermostable filler (talc) was used which enabled lower temperature printing around 110 °C and the stability of the used APIs was not affected [39].

In the same year, Sadia et al. decided to use a pharmaceutical-grade non-melting filler (TCP) through the hot-melt extrusion to allow a consistent flow from the nozzle of the printer. This novel approach meant the addition of 20–50% non-melting component to the filament and four model drugs were incorporated separately. This process aimed at the fabrication of well-defined caplets. In the case of 5-ASA and prednisolone, 93% of the drug contents remained intact in the tablet but a significant drop in captopril content was observed due to thermal degradation. This procedure made the manufacturing of personalized immediate release tablets easy [40]. In Figure 5. the most important research results can be seen between 1996–2016.

Acosta-Vélez reported the production of a biocompatible photocurable pharmaceutical polymer for inkjet 3D printing that was suitable for the manufacturing of hydrophilic active pharmaceutical ingredients. More specifically, hyaluronic acid was functionalized with norbornene moieties. This conjugate in the presence of poly(ethylene) glycol dithiol, Eosin Y, and a visible light source underwent a polymerization reaction. The manufactured bioink was loaded with ropinirole HCL and dispensed through a piezoelectric nozzle onto a blank preform tablet, and then polymerized. The study confirmed the potential of inkjet printing for the rapid production of tablets through the deposition of a photocurable bioink designed for hydrophilic APIs [43].

Beck et al. combined two important, innovative fields: additive manufacturing and nanotechnology. The researchers’ idea was to first create their own filament with a channeling agent with hot-melt extrusion then FDM print it and finally load the previously fabricated channels with nanocapsules by soaking. The researchers believed that this method could improve the delivery of the drugs [44].

Chai et al. planned an intragastric floating tablet. For the manufacturing, first domperidone was hot-melt extruded with hydroxypropyl cellulose and then the produced filament was FDM 3D printed. Based on the authors’ findings, the sample with a hollow structure was successfully fabricated and the buoyancy of tablets was closely related to their densities. Due to the rigid shells produced by the melting deposition, HPC polymer chains dissociated slowly [45].

An oral dual-compartmental dosage unit was designed for the treatment of tuberculosis (Figure 6). The aim of the research was to physically isolate and modulate the release profile of an anti-tuberculosis drug combination because rifampicin and isoniazid negatively interact with each other upon simultaneous release in an acidic environment. The samples were fabricated in two steps; first, 3D printing of the outer structure, followed by hot-melt extrusion of the two different drug-containing filaments. This way, the two APIs were separated and resulted in modified release and an effective retardation, based on the authors’ findings [46].

A DuoTablet was designed, which meant that for the first time glipizide was hot-melt extruded with PVA and then the drug-loaded filament was printed and formed a double-chamber device composed of a tablet embedded within a larger tablet [47].

Gyroid lattice printlets were designed containing paracetamol with SLS technology. The novel structure was able to modulate the drug release from all four polymers. This work was the first to demonstrate the feasibility of using SLS to achieve customized drug release properties of several polymers, and avoided the alteration of the formulation composition [48].

The goal of Hollander et al. was to study the printability of poly(dimethyl siloxane) (PDMS) with a semi-solid extrusion printer in combination with the UV-assisted crosslinking technology using UV-LED light to produce drug delivery systems. Samples with different pore sizes and API amount were prepared and contained prednisolone as a model drug. By altering the surface area/volume ratio, it was possible to create structures with different release rates. The study shows that this 3D printing technique in combination with UV-LED crosslinking was an applicable method and an interesting alternative for manufacturing controlled release devices containing temperature-susceptible drugs [49].

Kollamaram et al. aimed to fabricate low-melting and thermolabile drugs by reducing the FDM printing temperature. For this purpose, two immediate release polymers, Kollidon VA64 and Kollidon 12PF, were investigated and ramipril was used as the model low melting point drug (109 °C). The drug loaded filaments were extruded at 70 °C and contained 3 *w*/*w*% API, while the printing temperature was 90 °C. This work demonstrated that the selection and use of new excipients could make this technique suitable for drugs with lower melting temperatures [50].

Gastro-floating tablet were created with three kinds of infill percentage and prepared by hydroxypropyl methylcellulose (HPMC K4M) and hydroxypropyl methylcellulose (HPMC E15) as hydrophilic matrices and microcrystalline cellulose (MCC PH101) as extrusion molding agent (Figure 7). The study determined that floating could be maintained for up to eight hours with the combination of traditional pharmaceutical excipients and a modern technique [51].

Tablets with a novel design approach of caplets with perforated channels were fabricated by Sadia et al. to accelerate drug release from FDM 3D-printed samples (Figure 8). The experimental arrangement was to use different channel widths, lengths, and alignments. Based on the results, the parameters should be carefully considered in addition to surface area when optimizing drug release from samples. The incorporation of short channels could be adopted in the patterns of dosage forms built from polymeric filaments [52].

Scoutaris et al. created indomethacin (as a model drug) containing PEG filaments, then FDM 3D printed to construct chewable tablets. The shapes of the samples were variable (lion, heart, and teddy bear) for improved patient compliance in the case of children. This research also demonstrated that 3D printing could be effectively used for the manufacturing of personalized medication in the field of pediatrics [53].

In the same year a high—up to 60 *w*/*w*%—API-containing filaments were designed by TPU polymers and a prolonged release profile was achieved with a time period of 24 h. High drug loaded filaments were fabricated which were heat stable through the FDM 3D printing [54]. In Figure 9. the most important research results can be seen between 2017–2018.

Goyanes et al. aimed to produce a drug delivery system for a rare metabolic disorder called maple syrup urine disease, which required strict dietary restriction and oral supplementation of isoleucine. In the research, isoleucine containing printlets were constructed in six diverse flavors and four distinct API amounts with a special 3D printer (The Magic Candy Factory). The dissolution profile of the samples was adequate and the patients—with different preferences in terms of flavor and color—reported good acceptability of the formulations [55].

Öblom et al. designed isoniazid-containing filaments and then printed tablets with FDM printing technology. As a polymer, nonidentical cellulose based (such as HPMC, HPC, or Eudragit) filaments were fabricated with a constant content of 30 *w*/*w*% API. The effect of the used polymer, the size, and the infill percentage were investigated using the dissolution profile. Drug release characteristics could be altered by changing these critical printing parameters and allowing personalization of the tablets [56].

A polypill with SLA technique was manufactured from six different APIs: paracetamol, chloramphenicol, acetylsalicylic acid, naproxen, caffeine, and prednisolone, where the created structure was cylinder (Figure 10) or ring shaped. For the fabrication, a novel method was developed to fabricate multi-layered constructs with variable drug contents and shapes [57].

Nineteen semisolid formulations were prepared for a fractional factorial design. The variables were the amount of the API and unalike soluble and insoluble excipients. First, a Carbopol gel was made; then, with diclofenac sodium a semisolid pasta was created and then this special “ink” was 3D printed with a Bioplotter printer. The researchers found out that with the determination of critical process parameters a robust and consistent 3D printing method could be achieved [58].

Awad et al. decided to manufacture tablets by 3D printing technology with braille and moon patterns in various shapes for patients with visual impairment. Printlets with different shapes were fabricated to offer additional information, such as the medication indication or its dosing regimen. Despite the presence of the patterns, the printlets retained their original properties of a conventional tablet [59].

A special technique was used for the avoidance of high printing temperature in the case of FDM printing. A study was performed to develop novel core-shell gastroretentive floating pulsatile drug delivery systems using a hot-melt extrusion-paired FDM 3D printing and direct compression method. In the research, hydroxypropyl cellulose (HPC) and ethyl cellulose (EC)-based filaments were fabricated using hot-melt extrusion technology and were utilized as feedstock material for printing shells through the production. The directly compressed theophylline tablet was used as the core. The researchers fabricated a gastro-floating theophylline-containing drug delivery system [60].

An osmotically controlled dosage form was designed where the core contained the PVA, diltiazem, and osmogene. The 3D-printed core was covered by cellulose acetate to provide modified release. In the graphic, an imported hole and several linear cavities were shaped to achieve the controlled release profile [61].

Karavasili et al. created chocolate and corn syrup ink to print ibuprofen and paracetamol-containing dosage forms for children. The main concept was to fabricate a chewable tablet for pediatric use and to 3D print both hydrophilic and lipophilic APIs [62].

Melting solidification printing was used as a novel technique for the manufacture of oral solid dosage forms to avoid the use of solvents and high temperatures. This process was performed with a special ink—Gelucire^®^ 50/13 (fatty polyethylene glycol esters)—which could be used to obtain a floating sustained release system with improved dissolution and absorption of drugs, for example, from ricobendazole, which had showed a low and erratic bioavailability [63].

Tsintavi et al. dedicated their work to partially coating tablets with a glyceride, namely Precirol ATO 5, using a semi-solid 3D printer as an approach for tuning the release of two APIs, the hydrophilic methyl-levodopa hydrochloride and the lipophilic acyclovir. The surface coating percentage, the number of coating layers, and the coated sides of the tablet controlled the release profile and diverse dissolution profiles were reached [64].

Multi-layered polyprintlets were produced from PEG 300, PEGDA, and four different antihypertensive drugs: irbesartan, atenolol, hydrochlorothiazide, and amlodipine by SLA 3D printing. The created drug delivery system could deliver a low-dose combination therapy, but an interaction occurred between PEGDA and amlodipine. This unexpected drug–polymer interaction had a serious impact because it highlighted the need to screen the biocompatibility properties of photoreactive monomers to ensure the safety and compatibility of drug-loaded oral dosage forms produced by SLA [65].

Dapagliflozin-containing self-nanoemulsifying tablets were manufactured by semisolid pressure-assisted microsyringe (PAM) extrusion-based 3D printing technique. This work combined two very investigated fields recently in pharmaceutic manufacturing: SNEDDS and 3D printing. For the manufacturing, a liquid and a solid phase were fabricated separately. First, the solid phase was melted then the liquid phase ingredients were added. This semi-solid syringe was transferred to the extruder syringe when 3D printing could take place. This manufacturing enabled the manufacturing of a special drug delivery system with the combination of two innovative research fields [66].

A special method was designed for the fabrication of tablets with customizable dosages, durations, and combinations of multiple drugs by FDM 3D printing technology. The method and the structure of the tablet was simple: first, a template was printed by FDM 3Dprinter; then, a solution was poured into a PDMS mold where solidification take place. Finally, the samples were covered with pre-printed white wax coatings. The tablets were customized by varying the amount of excipient used, the height of the tablet, and the number and amount of used APIs (paracetamol, phenylephrine HCl, and diphenhydramine HCl). Based on the authors’ findings, with the use of templates a high variety of tablets could be constructed [67].

In a study, photoabsorbers were used to improve the SLS printability of five different colorless drugs and distinct excipients with low glass transition temperature and low stability. The forming mechanism of amorphous and crystalline polymers was sintering and melting, respectively. Immediate-release tablets with a high drug loading of 90% and sustained-release tablets with tunable dissolution behavior were successfully prepared, suggesting that the SLS technique had great prospects in producing personalized oral preparations [40,68]. In Figure 11, the most important research results could be seen since 2019.

In recent decades, the number of publications on 3D-printed tablets has multiplied. The above-described research was highlighted because of their novelty in some way. Since it would be completely impossible to characterize all the innovative research, we tempted to summarize even more research in the two tables below. The studies were classified based on the publication year and then in alphabetical order of the author. Table 1 summarizes the publications between 1996 and 2016, and Table 2 from 2017 to present.

Some advantages and limitations should also be mentioned. One of the biggest advantages against the conventional tablets were the possibility of incorporating several drug substances into one product to produce a polypill, which is personalized regarding both the combination of drug substances and the doses. These tablets would benefit the drug treatments of several medical conditions and would improve adherence to medications. We had to mention that low printing efficiency was one of the major limitations [69]. The other limitations were the lower productivity, higher costs, and incapability of production and delivery on-demand compared to the conventional tablets [70]. In addition, the healthcare professionals expressed some concerns associated with medication safety and quality aspects, including dose accuracy, quality control, stability, shelf life of formulations, and the identification of drug products at hospital wards [71]. Another research group named poor surface quality and mechanical strength of the final object as a limiting factor [72]. The possibility to 3D print personalized medications not only at an industry or pharmacy setting, nor compounding or community, but also even at the patient’s home could revolutionize the healthcare system [70].

**Table 1 pharmaceutics-14-01312-t001:** The grouping of the manufactured tablets based on the publication year and then in alphabetical order between 1996 and 2016.

Year	Type of 3D Printing	Type of Polymer	Type of API	Article
1996	desktop 3D printer	PCL, PEO	yellow and blue dye	Wu et al. [25]
2000	droplet binding	methacrylate copolymers	chlorpheniramine	Katstra et al. [26]
	droplet binding	methacrylate copolymers	chlorpheniramine, diclofenac	Rowe et al. [73]
2003	droplet binding (TheriForm™ process)	none (mannitol)	captopril	Lee et al. [27]
2006	droplet binding (TheriForm™ process)	Kollidon SR (80% polyvinyl acetate, 19% polyvinyl pyrrolidone)	pseudoephedrine	Wang et al. [28]
2007	bioceramic powder printing	Resomer RG502H (polylactide-polyglycolide 50:50)	vancomycin, ofloxacin, and tetracycline	Gbureck et al. [29]
	powder binding desktop 3D machine	PVP	acetaminophen	Yu et al. [30]
2009	powder binding desktop 3D machine	PVP K30	acetaminophen	Yu et al. [31]
2012	SLS	PCL	progesterone	Salmoria et al. [74]
2014	FDM	PVA	fluorescein	Goyanes et al. [75]
	Extrusion-based 3D printer (Fab@Home)	PAA	guaifenesin	Khaled et al. [32]
*2015—FDA approved*	*ZipDose*	*unknown*	*levetiracetam*	*Aprecia Pharmaceuticals* [33]
2015	FDM	PVA	paracetamol, caffeine	Goyanes et al. [35]
	FDM	PVA	paracetamol	Goyanes et al. [41]
	FDM	PVA	budesonide	Goyanes et al. [42]
	FDM	PVA	5- and 4- amino salicylic acid	Goyanes et al. [34]
	RegenHU 3D printer	HPMC	nifedipine, captopril, glipizide	Khaled et al. [36]
	RegenHU 3D printer	HPMC	ASA, HCT, atenolol, pravastatin, captopril	Khaled et al. [37]
	FDM	Eudragit RL100 and RS100, HPC	theophylline	Pietrzak et al. [76]
	FDM	PVA	prednisolone	Skowyra et al. [77]
2016	FDM	Eudragit EPO, Soluplus and PVA	felodipine	Alhijjaj et al. [78]
	FDM	PLA, HPMC	nitrofurantoin	Boetker et al. [38]
	FDM	PVA	paracetamol, caffeine	Goyanes et al. [79]
	FDM	PVP	dipyridamole or theophylline	Okwuosa et al. [39]
	FDM	Eudragit EPO	theophylline, 5-ASA, captopril, prednisolone	Sadia et al. [40]
	SLA	PEGDA	4-ASA, paracetamol	Wang et al. [17]

**Table 2 pharmaceutics-14-01312-t002:** The grouping of the manufactured tablets based on the publication year and then in alphabetical order between 2017 and 2021.

Year	Type of 3D Printing	Type of Polymer	Type of API	Article
2017	inkjet printing	PEG	ropinirole	Acosta-Vélez et al. [43]
	FDM	PCL, Eudragit RL 100	nanocapsules	Beck et al. [44]
	FDM	HPC	domperidone	Chai et al. [45]
	inkjet printing	PEGDA	ropinirole	Clark et al. [80]
	SLS	Kollicoat IR	paracetamol	Fina et al. [81]
	FDM	PEO, PLA	rifampicin, isoniazid	Genina et al. [46]
	FDM	HPMCAS	paracetamol	Goyanes et al. [82]
	FDM	HEC	food coloring	Goyanes et al. [83]
	FDM	beeswax	fenofibrate	Kyobula et al. [84]
	FDM	PVA	glipizide	Li et al. [47]
	FDM	PVP	theophylline	Okwuosa et al. [85]
	FDM	Kollidon^®^ VA64, Kollicoat^®^ IR, Affinsiol™15 cP and HPMCAS	haloperidol	Solanki et al. [86]
	FDM	PVA	curcumin	Tagami et al. [87]
	FDM	PLA	acetaminophen	Zhang et al. [88]
2018	inkjet printing with piezoelectric nozzle	PEG, PEGDA	naproxen	Acosta-Vélez et al. [89]
	FDM	HPC	theophylline	Arafat et al. [90]
	FDM	Eudrgait EPO	warfarin	Arafat et al. [91]
	SLS	Eudragit (L100-55 and RL)	paracetamol	Fina et al. [48]
	SLS	HPMC E5, Kollidon VA64	paracetamol	Fina et al. [92]
	UV-assisted crosslinking technology	PDMS	prednisolone	Hollander et al. [49]
	ZMorph^®^	Kollicoat^®^, PLA	aripiprazole	Jamróz et al. [69]
	RegenHU bioprinter	PVP K25	paracetamol	Khaled et al. [93]
	FDM	Kollidon VA64, Kollidon 12PF	ramipril	Kollamaram et al. [50]
	extrusion-based MAMII	HPMC K4M, HPMC E15, MCC PH101, PVP	dipyridamole	Li et al. [51]
	SLA	PEGda	paracetamol	Robles- Martinez et al. [94]
	SLS	HPMC	paracetamol	Trenfield et al. [95]
	FDM	Polyplasdone-XL^®^	hydrochlorothiazide	Sadia et al. [52]
	FDM	PEG	indomethacin	Scoutaris et al. [53]
	FDM	TPU	theophylline, metformin	Verstraete et al. [54]
2019	SLS	Kollidon^®^ VA 64	diclofenac	Barakh Ali et al. [96]
	direct single-screw powder extruder (FabRx)	HPC	itraconazole	Goyanes et al. [97]
	specially adapted 3D printer (The Magic Candy Factory)	pectin	isoleucine	Goyanes et al. [55]
	FDM	HPMC	carvedilol	Ilyés et al. [98]
	FDM	PEO	theophylline	Isreb et al. [99]
	FDM	Eudragit^®^ RS 100	acetaminophen	Krause et al. [100]
	FDM	HPMCAS, PEG 400	pregabalin	Lamichhane et al. [101]
	FDM	Cellulose based polymers	isoniazid	Öblom et al. [56]
	SLA	PEGda	paracetamol, chloramphenicol, acetylsalicylic acid, naproxen, caffeine, prednisolone	Robles-Martinez et al. [57]
	FDM	HPMC	acyclovir	Shin et al. [102]
	Bioplotter 3D printer	Polyplasdone	diclofenac sodium	Zidan et al. [58]
	pressure-assisted microsyringe	PVP	ginkgolide	Wen et al. [103]
	FDM	PVA	paracetamol	Xu et al. [104]
2020	SLS	Kollidon VA64	ondansetron	Allahham et al. [105]
	SLS	Kollidon VA64	paracetamol	Awad et al. [59]
	FDM	HPMC	theophylline	Cheng et al. [106]
	semi-solid 3D extrusion printer	HPCM	levetiracetam	Cui et al. [72]
	FDM	HPC, EC	theophylline	Dumpa et al. [60]
	FDM	HPC	caffeine	Fanous et al. [107]
	FDM	PVA	diltiazem	Gioumouxouzis et al. [61]
	SLS	Kollicoat^®^ IR	lopinavir	Hamed et al. [108]
	FDM	Kollicoat^®^ IR, PLA, PVA	bicalutamide	Jamróz et al. [109]
	DLP, SLS, SSE, FDM	PVA, PEGDA	placebo	Januskaite et al. [110]
	inkjet technology XYZprinting 3D Food Printer (Model 3C10A)	chocolate, corn syrup	ibuprofen, paracetamol	Karavasili et al. [62]
	SLS	MCC	clindamycin	Mohamed et al. [111]
	direct powder extrusion	PEO	tramadol	Ong et al. [112]
	melting solidification printing process	Gelucire 50/13	ricobendazole	Real et al. [63]
	semi-solid 3D printer	Precirol ATO 5	methyldopa, acyclovir	Tsintavi et al. [64]
	FDM	HPC	cinnarizine	Vo et al. [113]
	SLA	PEG 300, PEGDA	irbesartan, atenolol, hydrochlorothiazide, amlodipine	Wu et al. [65]
2021	pressure-assisted microsyringe	PEG 400, PEG 6000	dapagliflozin	Algahtani et al. [66]
	direct powder extrusion	Kollidon VA64	praziquantel	Boniatti et al. [114]
	SLS	PVPA	ropinirole	Davis et al. [115]
	SSE	emulsion gel	fenofibrate	Johannesson et al. [116]
	FDM	PEG 1000	paracetamol, phenylephrine HCl, diphenhydramine HCl	Tan et al. [67]
	FDM	PCL	indomethacin, theophylline	Viidik et al. [117]
	FDM	PEGDA	warfarin sodium	Xu et al. [118]
	SLS	PVA	indomethacin, nifedipine, tinidazole, ibuprofen, metoprolol, paracetamol, diclofenac sodium	Yang et al. [68]

### 2.2. Capsules

The first 3D-printed capsular devices were manufactured in 2015 by Melocchi et al. For the manufacturing, hydroxypropyl cellulose-containing filaments were created by hot-melt extrusion and then the filament was 3D printed. The manufactured samples were swellable erodible capsules for oral pulsatile drug release [119].

Fused deposition modeling and inkjet printing were used to fabricate capsules from different polymer formulations. The capsules were formed by three parts: two hollow parts which had a cylindrical closed end and a rounded open end; the middle part acted like a joint and a partition (Figure 12). The hollow parts differed in geometry and wall-thickness. The samples were filled with model APIs and the results showed that the device was able to successfully release the model APIs in pulses within 2 h [120].

A research group combined the versatility of 3D printing capsules with controlled geometry and the drug release properties of nanocellulose hydrogel to accurately modulate its drug release properties. As a novel method, the capsules were filled with a drug dispersion composed of model compounds and anionic cellulose nanofiber hydrogel. The main benefits of this device were that the release could be modulated simply by modulating the inner geometry of the PLA capsule and as the API did not undergo heating a wide range of APIs could be used. e.g., proteins and liposomes [121].

As it could be seen. capsules were investigated by a few research groups because only hard-shell capsules can be manufactured by 3D printing. The advantage of the manufacturing by 3D printing, more or less the same as in the case of the tablets as personalized drug delivery systems, is that it could be made with flexible on-demand doses with better health outcomes. As a limitation, a research group mentioned the API stability and the low amount of pharmaceutical grade polymeric carriers [119]. The published research on the created capsules can be found in Table 3.

### 2.3. Orodispersible Films

The first 3D-printed oral film was printed by thermal inkjet printing where the used API (salbutamol sulfate) was dissolved in the aqueous solution, the ink cartridges were filled with this solution, and it was printed onto a commercial potato starch film. The authors concluded that this process was suitable for the manufacturing of aqueous drug solutions into thin polymer films but the viscosity and API stability had to be controlled [125].

In another work, the aim was to evaluate the applicability of orodispersible films (ODFs), porous copy paper sheets, and water impermeable transparency films (TFs) which contained rasagiline mesylate (RM) as a low dose active pharmaceutical ingredient. Flexible doses of the API were obtained by printing several subsequent layers on top of the already printed ones, using an off-the-shelf consumer thermal inkjet (TIJ) printer [126].

A research group manufactured the drug dosage form with a special 3D printing method which incorporated two different methods: piezoelectric- and solenoid valve-based inkjet printing technologies to allow the dispensing of an extensive range of fluids. The research demonstrated the opportunity to 3D print a wide range of formulations for the patient needs. The fabrication avoided the risk of drug degradation by ink heating and of substrate damage (by contact printing) and the manufacturing scheme avoided the emergence of defects [127].

Vakili et al. used inkjet printing to create orodispersible films, which contained propranolol hydrochloride. The drug delivery systems were designed with escalating doses of propranolol hydrochloride on three different substrates and three unalike area sizes were used through the 3D printing with thermal inkjet printing technology. A thin sweetener coating layer of saccharin was successfully included in the final dosage form to increase the patient compliance among pediatric patients [128].

Aripiprazole-containing orodispersible films were fabricated with FDM technology from PVA by Jamróz et al. The aripiprazole in the sample is fully amorphous due to the two-step hot-melt extrusion process (filament fabrication and 3D printing) and the high concentration of PVA polymer helped to maintain the amorphous form [129].

In a study, benzydamine hydrochloride and HEC were used for the manufacturing of a printing dispersion. For the 3D printing, a modified FDM technique was used in which the FDM extruder was replaced by linear syringe pump. This method could be implemented into compounding practice of pharmacies allowing preparation of ODFs for the patients in small batches, and could eliminate preliminary test printing [130].

A research group aimed to investigate semisolid extrusion 3D printing for production of warfarin-containing orodispersible films. The applied 3D printing method was unique because a one-step-process utilized disposable syringes hindering the printing material to be in contact with the printing equipment. The method was successfully utilized to produce transparent, smooth and thin, yet flexible and strong orodispersible films containing therapeutic doses of warfarin. The authors found it to be a potential method for on-demand compounding right at the bedside [131].

With the 3D printing technique, the limitations of the conventional manufacturing process could be eliminated. In addition, a precise amount of API could be printed, and it was feasible to print fixed API combinations. These attributes made the oromucosal films especially interesting for the administration of potent APIs, for example, for the treatment of cardiovascular disorders, schizophrenia, or migraine. A huge advantage of the printing technique was the possibility to integrate safety features in the form of QR codes with the dosage form. However, some challenges such as increased dosing remained even with the use of 3D printing [132]. Some other limitations were the dose inaccuracy and the liquid formulation by artificial instruments based on a research group [133]. The published research on the fabricated orodispersible films can be found in Table 4.

### 2.4. Implants

Levofloxacin-containing PLA implants were designed with inkjet printing. The manufactured samples were 10 mm in width and rounded. In this research, a complex release profile was demonstrated in the 100-day monitoring period when one pulse of release appeared from the 5th to 25th day, and another pulse began at the 50th day and ended at the 80th day, with a lag time of 25 days between the two pulses, wherein a steady state of release was observed at about 5 µg/mL [141].

Rifampicin and isoniazid-containing multi-layered concentric cylindrical implants were fabricated against tuberculosis. The multi-layered concentric cylinder was divided into four layers from the center to the periphery and the APIs were distributed individually into the different layers in a specific sequence of isoniazid–rifampicin–isoniazid–rifampicin. The dissolution tests proved that the API liberation takes place orderly from the outside to the center and the peak concentrations were between 8 and 12 days. In this study, a programmed release multi-drug implant with a complex construction was fabricated by 3D printing [142].

A research group prepared dexamethasone-containing tailored drug delivery platforms where two distinct designs—structure A: rolled and sealed; structure B: layer-by-layer—were extrusion printed. As the API liberation was continuous for more than 4 months, these samples could be used as implants [143].

Genina et al. manufactured intrauterine device and subcutaneous rods from ethylene vinyl acetate (EVA) copolymer with FDM printing. The samples were containing indomethacin as a model API and with the device the drug dissolution was over 30 days. A long-acting 3D-printed implantable system was built [144].

In a study, levofloxacin and tobramycin-containing implants were fabricated for the treatment of osteomyelitis. A multi-layered concentric cylinder construction was created by powder-based inkjet printing. A sustained and programmed drug delivery system was provided [145].

In a study, the effect of the used polymers on the drug release profile of quinine was examined as a model drug. The used polymers were Eudragit^®^ RS, PCL, PLLA, and EC and affected the dissolution profile of the samples. The fastest relative drug release was observed from PCL where the dissolved API amount was approximately 76% in 51 days and the lowest from Eudragit RS and EC with less than 5% of quinine release in 78 and 100 days, respectively [146].

Qamar et al. manufactured an implantable mesh for the treatment of hernia. PP and PVA meshes were produced with distinct pore size, shape, and thread thickness. The meshes were filled with ciprofloxacin for the management of hernia. Based on the research, animals implanted with ciprofloxacin HCl loaded meshes exhibited fewer fluctuations in body temperature and faster wound healing [147].

The purpose of a study was to demonstrate the applicability of 3D printing methods for the fabrication of patient-specific fixation implants that allow localized drug delivery. The 3D printing was used to fabricate gentamicin and methotrexate loaded fixation devices, including screws, pins, and bone plates [148].

In a study, PLLA samples were printed using a special Zcorp Zprinter 650 then immersed into the solution of various anticancer drugs such as cisplatin, ifosfamid, methotrexate, or doxorubicin and finally dried. The proposed 3D-printed drug delivery system could simultaneously realize individual local chemotherapy, multi-drug delivery, long-term sustainable drug release, and non-reoperation in osteosarcoma treatment [149].

Ciprofloxacin containing PLA implants were fabricated with the combination of semi-solid extrusion and fused-deposition modeling for the treatment of bone infections. The authors found this method more adequate than the conventional method for manufacturing [150].

The 3D printing technologies have promising potential for the manufacture of sophisticated drug implants and patient-specific macro-porosity implants with personalized drug release behavior. In general, the currently commercially available implants lack the personalization of the treatment and consideration of several issues such as anatomical differences, age, genders, and disease condition [151]. A research group determined that 3D-printed implants could display complex drug release patterns compared to conventionally fabricated drug implants [141]. The manufacturing of 3D printing provided the possibility to produce smaller, less invasive, and more site-specific implants compared to the conventional ones [152]. Even though the clinical data—to support the routine use of the 3D-printed customized implants—are currently limited, the patient-specific implants may gain more attention and popularity in the prophylaxis and treatment of diseases such as complicated bone infections and bone tuberculosis in the near future when the digital and manufacturing technology advances further [150].

Some published research on the produced implants can be found in Table 5.

### 2.5. TTS

Anti-acne drug loaded masks/patches were fabricated by Goyanes et al., but as this system provides transdermal delivery, we decided to subgroup the research here. In the research, salicylic-acid-containing filaments were used for the FDM 3D printing but the API showed significant thermal degradation. The manufacturing by SLA contained a higher amount of drug and showed no drug degradation, so the researchers found this method more adequate [160].

Yi et al. manufactured a 3D-printed biodegradable patch with a versatile shape and incorporated a high drug concentration for the achievement of a controlled drug release profile. The patches composed of poly(lactide-co-glycolide), polycaprolactone, and 5-fluorouracil were the antitumor agent. With the use of 3D printing technology, the geometry of the patch and the drug release kinetics could be manipulated. The patches were flexible, and released the drug over four weeks with minimized side effects [161].

A research group developed an electro hydrodynamic (EHD) printing technique to fabricate antibiotic-containing patches using polycaprolactone (PCL), polyvinyl pyrrolidone (PVP), and their composite system (PVP-PCL). Drug loaded 3D patches possessed perfectly aligned fibers giving rise to fibrous strut orientation, variable inter-strut pore size, and controlled film width (via layering). The used polymer type and the printed patch void size affected the dissolution profiles [162].

If we compared the advantages and the limitations, we found that the conventional manufacturing process involves solvent evaporation and multiple steps which were often time consuming in comparison with the 3D printing method. In case of the TTSs, the adhesion was a critical point and the skin could be undulating and curved, especially on the nose or head, which the conventional TTSs could not take into consideration [151]. The combination of 3D scanning and 3D printing had the potential to create personalized drug loaded devices, adapted in shape and size to individual patients [160]. The 3D printing of differently shaped patches demonstrated the ability to manipulate the drug release by altering the surface area which allowed precise control of the liberation amount and placement of the API, high efficacy, and minimized systemic toxicity [161]. A few published studies on the created TTSs can be found in Table 6.

### 2.6. Microneedles

In the early research of Ovsianikov et al., a placebo microneedle was developed by femtosecond laser two photon polymerization 3D printing technology [164]. In 2013, an amphotericin B containing microneedle was created by the combination of visible light dynamic mask micro stereolithography, micro molding, and piezoelectric inkjet printing. Based on the results, the printing process was found to be a scalable approach that could be used to incorporate pharmacologic agents, even with complex solubility profiles, within microneedles [165]. The same researchers fabricated miconazole-containing microneedles with the same technology but with a special polymer called Gantrez^®^ AN 169 BF (poly(methyl vinyl ether-co-maleic anhydride)). The manufactured sample had potential use in transdermal treatment of cutaneous fungal infections [166].

A dacarbazine-containing drug delivery system was produced which could be used for the therapy of skin cancer locally. For the manufacturing, a special 3D printing process was used named as multi-material micro stereolithography (μSL). First, the microneedle array was built and then the API was added with blending due to the crosslinking effect of the polymer (Figure 13) [167].

Insulin polymeric layers on metal microneedles were constructed by Ross et al. The dissolution profiles showed rapid insulin release rates in the first 20 min, suggesting that solid-state insulin delivery via microneedles was feasible [168].

Lim et al. used 3D printing for manufacturing a dual-function microneedle array on personalized curved surfaces (microneedle splint) for drug delivery and splinting of the affected finger. Sufficient penetration efficiency was achieved, and the final microneedle splint showed biocompatibility. A significantly higher amount of diclofenac permeated through the skin with the use of the microneedle splint as compared to intact skin. The fabricated microneedle splint could thus be a potential new approach to treat trigger finger via personalized splinting without affecting normal hand function [169].

Two research groups produced insulin-releasing microneedles from Dental SG with the use of inkjet printing and the SLA technique. In the case of the research of Pere et al., the API was released rapidly within 30 min [170]. Economidou et al. reached better glucose levels and hypoglycemia control [171].

One of the biggest advantages of 3D printing was that the technique allowed the incorporation of pharmacologic agents even with a complex solubility profile into the microneedles for patient needs [165]. The manufactured microneedles provided accurate dosing which also maintained the mechanical strength in comparison with the existing microneedles, e.g., made of metal. Furthermore, microneedles that have extremely fine tips in nano scale allowed good penetration of skin and potentially cell targeted delivery [172]. A few studies on the 3D-printed microneedles can be found in Table 7.

### 2.7. Vaginal Drug Delivery Systems

Even though this subsection consists of numerous drug delivery systems, we would like to discuss them together for easier accessibility.

The article of Genina et al. was already discussed in the section of implants because two distinct type of drug delivery systems were fabricated: intrauterine device (IUD) and subcutaneous rods [144].

Tappa et al. designed estrogen and progesterone-containing PCL filaments and FDM printed into different structures: mesh, IUD, and subdermal implant. As the FDM printing with PCL was possible at lower temperatures (around 130 °C), the API remained stable based on the TG/DSC results. This was the first study to offer data on hormone-loaded 3D-printed constructs [176].

Bioadhesive vaginal films were produced by Varan et al., where the used APIs were paclitaxel and cidofovir, which had antiviral efficacy and used for the treatment of cervical cancer locally. The basic idea was to increase the solubility of the paclitaxel with a cyclodextrin complex while cidofovir was encapsulated in PCL nanoparticles to manufacture an ink [177].

Our research group fabricated a vaginal ring as a carrier system for the treatment of bacterial vaginosis. For the manufacturing, FDM 3D printing was used and then these pre-printed samples were easily filled with the jellified APIs (vaginal gels) depending on the patient’s need. (Figure 14). This article focused on avoiding the loss and decomposition of API through the printing process and manufacturing a drug delivery system, which could be directly and rapidly printed at the bedside or in the pharmacy as individualized medication [178].

The currently available approaches did not consider the specific anatomy of each patient, the patient’s medical conditions, age, and gender specificity which limited the effective therapy whereas every patient is unique and should require different doses of drugs and hormones [179]. In addition, other factors such as variability in drug absorption related with menstrual cycle, menopause, and pregnancy were different among patients [180]. Therefore, patient-specific systems are needed to provide personalized shape, size, and tailored drug release to improve the efficacy and to increase compliance. The 3D-printed delivery system in this regard can provide personalized geometry, snug fit, increase efficacy, improve patient compliance, and prevent the post-surgical complications [151].

Some research on the manufacturing of vaginal drug delivery systems can be found in Table 8. The research was classified based on the publication year and then in alphabetical order. In the first column, the type of the drug delivery system is described.

### 2.8. Micro and Nanoscale Dosage Forms

Another area that concerns the researchers is the possible manufacturing of micro and nanoscale drug delivery systems. In a study, a poorly soluble model drug was used to construct a 10% folic acid containing nanosuspension and the printing was performed on an inkjet-based microdosing dispenser head. In the research, the authors found this method adequate for the incorporation of poorly soluble drugs to increase the oral absorption [183].

Scoutaris et al. produced poorly soluble felodipine and PVP-containing solution for inkjet printing. Based on the authors’ research, inkjet printing could be used to prepare this novel drug dosage form consisting of micro-sized dried deposits from sprayed picolitre droplets containing a drug on a substrate. The novelty of the work was that a scalable dosage form could be produced whereby many droplets could be produced to achieve a dissolution profile equivalent to conventional bulk dosage formulations [184].

In another study, rifampicin and PLGA-containing inks were fabricated for micro-pattern printing on a glass or titanium carrier. This research combines special ink formulations in microscale range and the author’s idea was to use these micro-patterns in orthopedic surgeries [185].

Paclitaxel-containing PLGA inks were created for piezoelectric inkjet printing where four different shapes were printed. Microparticles with diverse geometries exhibited non-similar drug release rates mainly due to nonidentical surface areas [186].

Some studies on the manufacturing of micro and nanoscale systems can be found in Table 9. The research was classified based on the publication year and then in alphabetical order. In the first column, the type of the drug delivery system was described.

### 2.9. Other

A few other interesting but not mandatory ‘drug delivery systems’ are discussed in this section. Kitson et al. used ibuprofen with a modified Prusa FDM 3D printer where the API was deposited in a pre-printed vessel. In this case, the API could be directly printed to a drug delivery system, e.g., tablet or capsule, to receive a ready-to-use drug dosage form [187].

Long et al. 3D-printed a chitosan-pectin hydrogel. To the gel, lidocaine hydrochloride was incorporated as a potential wound dressing candidate. The scaffolds were printed using an extrusion-based 3D printer using a mechanical positive displacement dispensing system followed by lyophilization. The research confirmed the possible manufacturing of hydrogels with 3D printing [188].

Seoane-Viaño et al. fabricated tacrolimus-containing rectal suppositories with semi-solid extrusion-based 3D printing. For the printing, Gelucire 44/14 and coconut oil were used. The designed drug dosage form could contribute to a long-lasting effect on the treatment of colitis [189,190].

## 3. Future Perspective

Three-dimensional printing enables the manufacture of personalized drug delivery systems. With the use of this modern technique, cost-effectiveness, simplified production, complex formulations, and increased opportunities for collaboration can be achieved. The high flexibility provides the production of a multitude of drug products with tailored release profiles and designs [20]. Additionally, 3D printing has to challenge some limitations in the future. As 3D printing right now cannot compete with the speed of the conventional drug manufacturing method, in the future the improvement in the printing speed and resolution has to be adjusted which can also reduce the energy consumption and cost as a recent, huge concern. The printed product dimension accuracy and scale size (e.g., micro or nanoscale) must be improved also. As 3D printing technology is limited in terms of materials, the manufacturing of novel polymers, inks, etc., are still required and necessary for the proper manufacturing [191]. Even though the FDA approved Spritam^®^ in 2015, the used ZipDose^®^ technology was similar to traditional powder compaction that could help the process of approval. At present, there are no fixed guidelines for the regulation of 3D-printed pharmaceutical dosage forms, so the researchers have to face regulatory concerns [151]. Another interesting area is bioprinting where living cells, tissues, or organs can be printed and exploratory studies of bioprinting in pharmaceutics have shown promising applications of this technique, for example, in the field of target identification and validation or in vitro efficacy assessment [19].

## 4. Conclusions

In conclusion, 3D printing represents a very interesting and modern technique in the field of drug manufacturing which is about to revolutionize the health industry. Three-dimensional printing is a layer-by-layer, automated process, which enables the manufacturing of complex, personalized products on-demand. In the last decade, the number of publications multiplied year-by-year. The FDA approved the first and only 3D-printed drug in 2015, which supported the commercial feasibility of this technology. In the last two decades, dozens of research groups aimed to manufacture different drug dosage forms, e.g., tablets, capsules, implants, even rectal suppositories by 3D printing to improve the safety, efficacy, and tolerability of medicines and provide individualized therapy for those most in need.

## Figures and Tables

**Figure 1 pharmaceutics-14-01312-f001:**
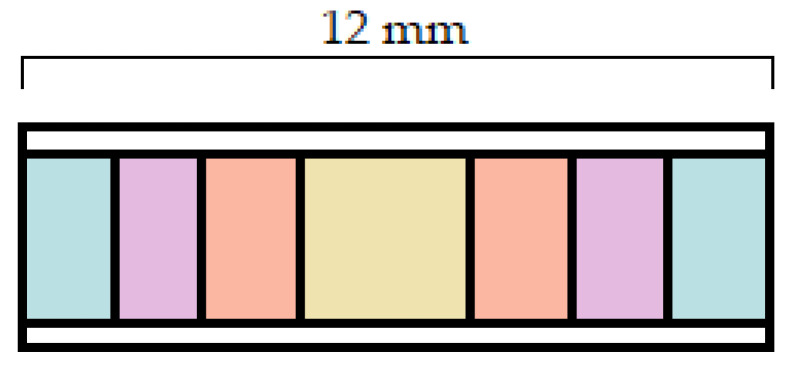
Cross-section of the acetaminophen-containing matrix tablets based on the authors’ figure. The different colors label dissimilar compartments [30].

**Figure 2 pharmaceutics-14-01312-f002:**
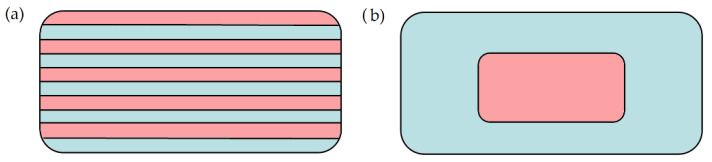
Cross-section of the constructed drug dosage forms in the article of Goyanes et al. (**a**) Sectioned multilayer tablet, (**b**) sectioned DuoCaplet (caplet in caplet) [35].

**Figure 3 pharmaceutics-14-01312-f003:**
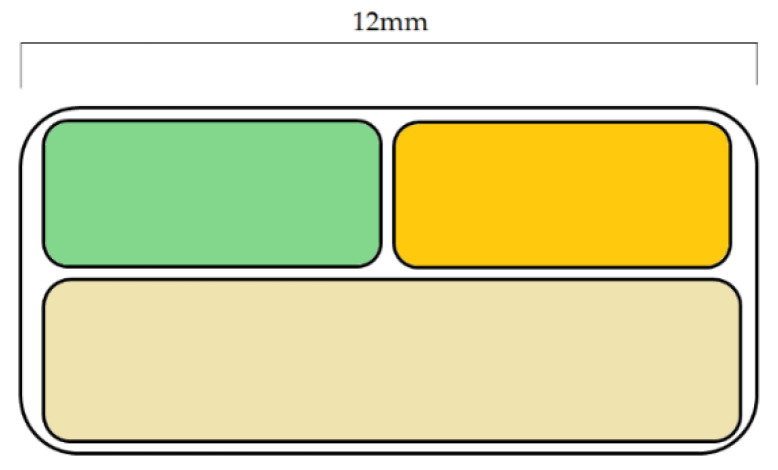
Cross-section of the printed polypills where three diverse compartments were created (labeled with three nonidentical colors) [36].

**Figure 4 pharmaceutics-14-01312-f004:**
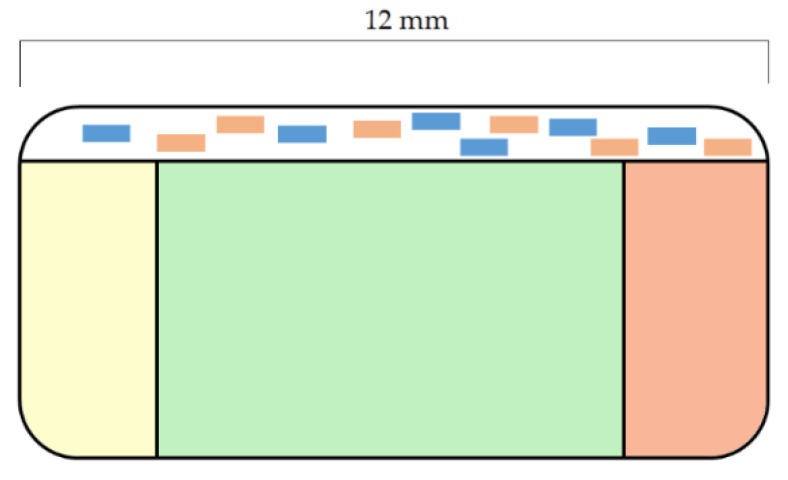
Cross-section of the polypills where ASA and HCT formulations were located in the upper immediate release compartments (labeled with blue and orange rectangles) and atenolol, pravastatin, and ramipril formulations were in three distinct extended release compartments (labeled with yellow, green, and peach blossom). The three compartments were the same size but could be visualized like this because of the original design of the “cake slice” [37].

**Figure 5 pharmaceutics-14-01312-f005:**
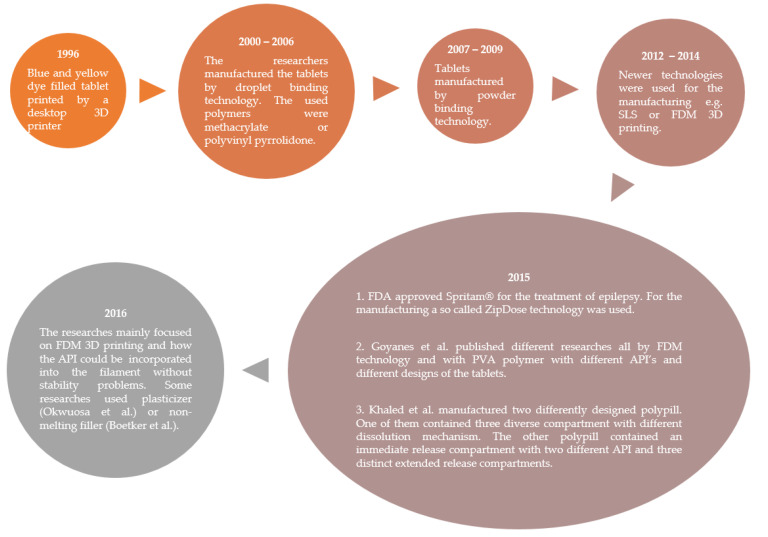
Flow chart on the described tablet manufacturing methods and main breakthroughs between 1996 and 2016 [34,35,36,37,39,40,41,42].

**Figure 6 pharmaceutics-14-01312-f006:**
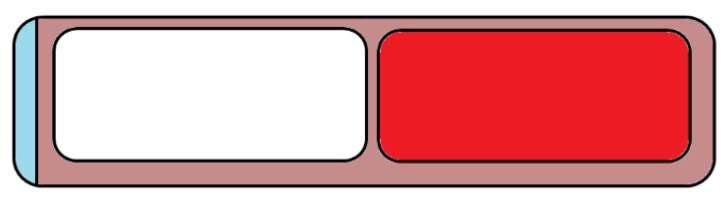
Cross-section of the dual-compartmental dosage form designed by Genina et al. In the research, isoniazid (white colored) and rifampicin (red colored) were hot-melt extruded and then 3D printed into the polymeric cap (brown colored) and closed with a cap (blue colored) [46].

**Figure 7 pharmaceutics-14-01312-f007:**
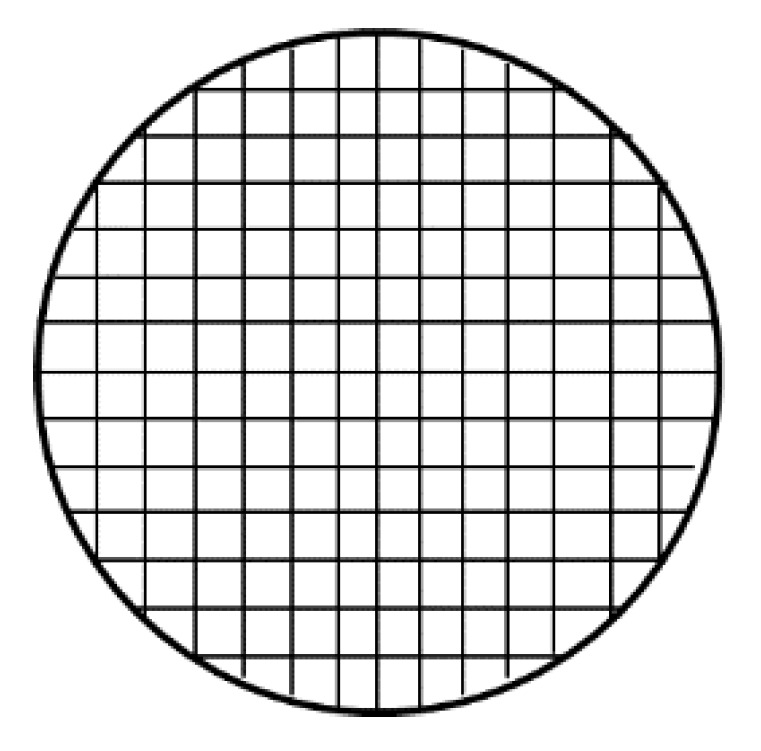
Cross-section of the 3D-printed gastro-floating tablets with 30% infill percentage rate [51].

**Figure 8 pharmaceutics-14-01312-f008:**
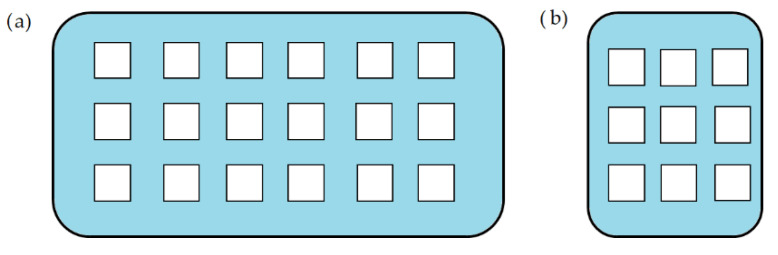
Cross-section of the 3D-printed channeled tablets. Each white square represents a channel. (**a**) Channels parallel to the longer side; (**b**) channels parallel to the shorter side [52].

**Figure 9 pharmaceutics-14-01312-f009:**
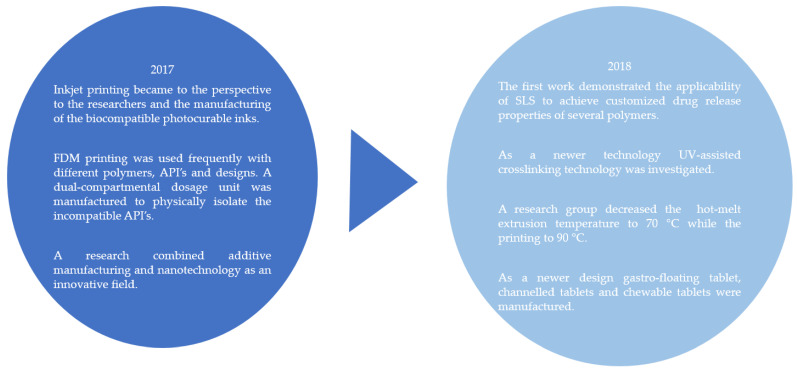
Flow chart of the described tablet manufacturing methods and main breakthroughs in 2017 and 2018.

**Figure 10 pharmaceutics-14-01312-f010:**
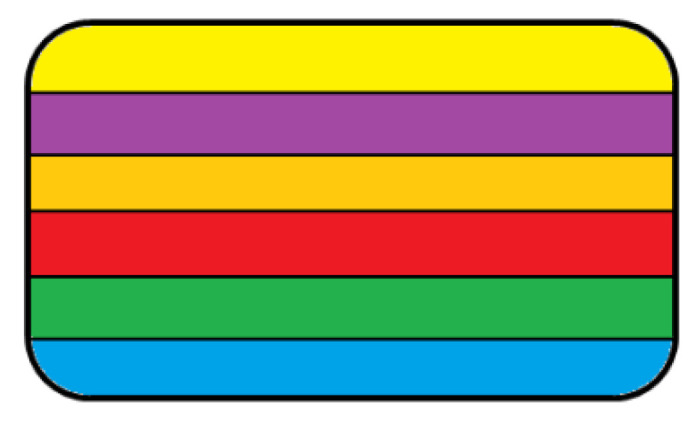
Cross-section of the cylinder-shaped polypill. Each color represents diverse API-containing layers: naproxen—yellow; aspirin—purple; paracetamol—orange; caffeine—red; chloramphenicol—green; and prednisolone—blue [57].

**Figure 11 pharmaceutics-14-01312-f011:**

Flow chart of the described tablet manufacturing methods and main breakthroughs since 2019.

**Figure 12 pharmaceutics-14-01312-f012:**
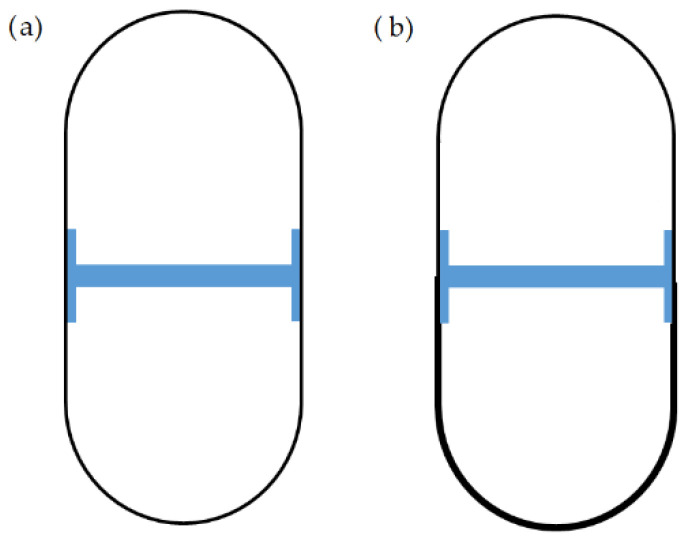
Cross-section of the designed capsules. (**a**) With the same wall thickness, (**b**) with different wall thicknesses [120].

**Figure 13 pharmaceutics-14-01312-f013:**
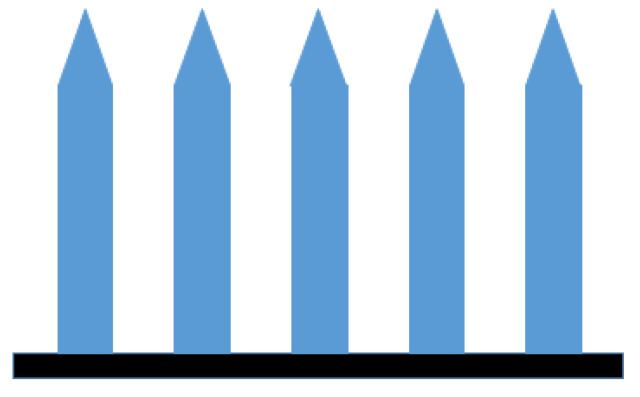
Cross-sectional design of the microneedle array [167].

**Figure 14 pharmaceutics-14-01312-f014:**
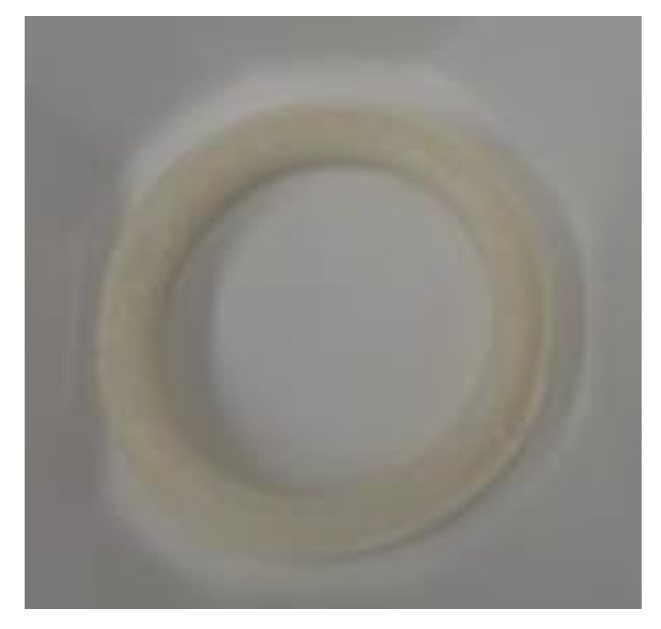
The designed vaginal drug delivery system by our research group [178].

**Table 3 pharmaceutics-14-01312-t003:** The grouping of the manufactured capsules based on the publication year and then in alphabetical order.

Year	Type of 3D Printing	Type of Polymer	Type of API	Article
2015	FDM	HPC	no (yellow and blue dye)	Melocchi et al. [119]
2016	FDM	PLA, EC, HPC, HPMC, HPMCAS, various Eugradit, PEO, PVA, Soluplus, PEG 400 and 8000	acetaminophen, furosemide	Melocchi et al. [122]
2017	FDM, Inkjet	PLA, PVA, polymer formulations	no (yellow and blue dye)	Maroni et al. [120]
2018	FDM	PVA-PEG, HPC, EC	Fluorodeoxyglucose (18F-FDG) TRACERlab MX synthesizer (GE Healthcare^®^)	Basit et al. [123]
	FDM	HPC, PLA	caffeine, blue and yellow dye	Melocchi et al. [124]
2020	FDM	PLA	metoprolol, nadalolol	Auviven et al. [121]

**Table 4 pharmaceutics-14-01312-t004:** The grouping of the manufactured orodispersible films based on the publication year and then in alphabetical order.

Year	Type of 3D Printing	Type of Polymer	Type of API	Article
2011	thermal inkjet printing	no need	salbutamol sulphate	Buanz et al. [125]
2012	inkjet and flexographic printing	EC	riboflavin, propranolol	Genina et al. [134]
2013	thermal inkjet printing	crospovidone (Kollidon CL-M)	rasagiline mesylate	Genina et al. [126]
2016	inkjet printing	PEGylated PLGA	sodium picosulphate	Planchette et al. [127]
	inkjet printing	HPC	propranolol hydrochloride	Vakili et al. [128]
2017	FDM	PVA	aripiprazole	Jamróz et al. [129]
2018	FDM	PVA, PEO, PEG	ibuprofen, paracetamol	Ehtezazi et al. [135]
2019	semi-solid extrusion	hydroxypropyl-β-cyclodextrin, cellulose	carbamazepine	Conceição et al. [136]
	FDM	PVA	diclofenac sodium	Eleftheriadis et al. [137]
	EXT, IJP	HPC	warfarin	Öblom et al. [138]
	Biobot	PVA	warfarin	Sjöholm et al. [131]
2020	modified FDM	maltodextrin, HEC	benzydamine hydrochloride	Elbl et al. [130]
	inkjet printing	PEO, HPC	prednisolone	Sjöholm et al. [139]
	semi-solid extrusion	HPMC	levocetirizine	Yan et al. [133]
2021	multitool 3D printer	HPMC	indomethacin	Germini et al. [140]

**Table 5 pharmaceutics-14-01312-t005:** The grouping of the manufactured implants based on the publication year and then in alphabetical order.

Year	Type of 3D Printing	Type of Polymer	Type of API	Article
2007	inkjet printing	L-PLA	levofloxacin	Huang et al. [141]
2009	inkjet printing	PDLLA	rifampicin, isoniazid	Wu et al. [142]
2012	extrusion based 3D printing	PLGA, PVA	dexamethasone	Rattanakit et al. [143]
2014	powder binding	PLLA	isoniazid	Wu et al. [153]
2015	3D-Bioplotter system	(3-hydroxybutyrate-co-3-hydroxyhexanoate) (PHBHHx)	isoniazid, rifampin	Min et al. [154]
	FDM (MakerBot^®^)	PLA	nitrofurantoin	Water et al. [155]
2016	FDM	PCL containing EVA	indomethacin	Genina et al. [144]
	inkjet powder printing	PDLLA	levofloxacin, tobramycin	Wu et al. [145]
2017	FDM	Eudragit RS, PCL, PLLA, EC	quinine	Kempin et al. [146]
2019	FDM, DMLS	PLA, PCL, titanium dioxide	doxycycline	Benmassaoud et al. [156]
	FDM	PP, PVA	ciprofloxacin	Qamar et al. [147]
	FDM	PLA	gentamicin, methotrexate	Tappa et al. [148]
2020	FDM	PLA, antibacterial PLA, PETG, PMMA	diclofenac sodium	Arany et al. [157]
	powder bed printing	PLLA	cisplatin, ifosfamid, methotrexate, doxorubicin	Wang et al. [149]
2021	SSE, FDM	PLA	ciprofloxacin	Cui et al. [150]
	SLA	Elastic Resin	lidocaine	Xu et al. [158]
	DLP	PEGDA, PEG 400	dexamethasone, phenyl bis phosphine oxide, β-carotene	Xu et al. [159]

**Table 6 pharmaceutics-14-01312-t006:** The grouping of some manufactured TTSs based on the publication year and then in alphabetical order.

Year	Type of 3D Printing	Type of Polymer	Type of API	Article
2016	FDM, SLA	Flex EcoPLA, PCL	salicylic acid	Goyanes et al. [160]
	inhouse extrusion-based 3D printer—multi-head deposition system	PLA, PCL	5-fluorouracil	Yi et al. [161]
2017	EHD	PCL, PCL/PVP	tetracycline hydrochloride	Wang et al. [162]
2021	FDM	PVP	quercetin	Chaudhari et al. [163]

**Table 7 pharmaceutics-14-01312-t007:** The grouping of some manufactured microneedles based on the publication year and then in alphabetical order.

Year	Type of 3D Printing	Type of Polymer	Type of API	Article
2007	femtosecond laser two photon polymerization	Ormocer^®^	none	Ovsianikov et al. [164]
2013	piezoelectric inkjet printing	PDMS, PMMA	amphotericin B	Boehm et al. [165]
2014	piezoelectric inkjet printing	Gantrez^®^ AN 169 BF (poly(methyl vinyl ether-co-maleic anhydride))	miconazole	Boehm et al. [166]
2015	multi-material microstereolithography (μSL)	poly(propylene fumarate)	dacarbazine	Lu et al. [167]
	inkjet printing	polyvinyl caprolactame-polyvinyl cetatepolyethylene glycol (SOL), poly(2-ethyl-2-oxazoline) (POX)	insulin	Ross et al. [168]
	inkjet printing	Soluplus^®^	5-fluorouracil	Uddin et al. [173]
2017	DLP	3DMCastable resin	diclofenac sodium	Lim et al. [169]
2018	SLA	medium viscosity alginate	blue dye, HepG2 cell encapsulation	Farias et al. [174]
	FDM	PLA	fluorescein	Luzuriaga et al. [175]
	inkjet printer	Dental SG	insulin	Pere et al. [170]
2019	SLA	Dental SG	insulin	Economidou et al. [171]

**Table 8 pharmaceutics-14-01312-t008:** The grouping of the manufactured vaginal drug delivery systems based on the publication year and then in alphabetical order.

Drug Delivery Type	Year	Type of 3D Printing	Type of Polymer	Type of API	Article
IUD, subcutaneous rod	2016	FDM	EVA, PCL	indomethacin	Genina et al. [144]
IUS		FDM	PCL	indomethacin	Holländer et al. [181]
Mesh, IUD, subdermal implant	2017	FDM	PCL	estrogen, progesterone	Tappa et al. [176]
Bioadhesive film		inkjet printing	PCL, PEG-PCL	paclitaxel, cidofovir	Varan et al. [177]
Bioadhesive film	2019	inkjet printing	PEG-PCL	paclitaxel, cidofovir	Varan et al. [182]
Intravaginal ring	2021	FDM	TPU	chloramphenicol, metronidazole	Arany et al. [178]

**Table 9 pharmaceutics-14-01312-t009:** The grouping of the manufactured micro- and nanoscale drug delivery systems based on the publication year and then in alphabetical order.

Drug Delivery Type	Year	Type of 3D Printing	Type of Polymer	Type of API	Article
Nanosuspension	2011	inkjet-based micro dosing dispenser head	none	folic acid	Pardeike et al. [183]
Micron-sized dried deposits		inkjet printing	PVP	felodipine	Scoutaris et al. [184]
Micropatterns	2012	inkjet printing	PLGA	rifampicin	Gu et al. [185]
Microparticles		piezoelectric inkjet printing	PLGA	paclitaxel	Lee et al. [186]

## Data Availability

Not applicable.

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
