# Peer review of "The Evolution of the 3D-Printed Drug Delivery Systems: A Review"

_pharmaceutics, 2022, doi:10.3390/pharmaceutics14071312_

Round 1
Reviewer 1 Report
1) A quite interesting read.
2) Spell check of text required (for example, line #102 "todemonstra", and there is many such typos).
3)In my opinion, if the authors can add a part to their review dedicated to the limitations and challenges faced by manufacturers of drug release systems in this area. This would greatly improve the readability and value of the review.
Author Response
Reply to Reviewer 1
Dear Reviewer,
We would like to thank you all for your constructive comments in the review. In this document, we tried to address the issues raised as best as possible. Our revision in the manuscript was highlighted by blue colour.
- A quite interesting read.
Thank you so much!
- Spell check of text required (for example, line #102 “todemonstra”, and there is many such typos).
Thank you for your comment, as it is a long review we tried to eliminate the spell mistakes as much as possible, but we would like to ask English editing for the thorough check.
- In my opinion, if the authors can add a part to their review dedicated to the limitations and challenges faced by manufacturers of drug release systems in this area. This would greatly improve the readability and value of the review.
Thank you for your question. New parts were inserted into the review. The advantages and limitations could be found at the end of every drug delivery system. A new section was made “3. Future perspective” which hopefully improved the readability and novelty of this work.

Reviewer 2 Report
The authors divided the 3D printing of drug delivery systems to tablets, capsules, orodispersible films, implants, transdermal delivery systems, microneedles, vaginal drug delivery systems and micro- and nanoscale dosage forms. Each category was fully discussed accoring to the period of the pulication finding. The idea is interesting and at the frontier of 3D printing. The authors seem to have collected enough literatures and they listed all the references with a brife dicussion.
The whole manuscript seems to be well written but the lack of integraty of each section need be revised. Several logical flow charts should be supplemented to make the paper structure more compact, which largely facilities the readers in understanding this manuscript. The content structure could be improved.
Author Response
Reply to Reviewer 2
Dear Reviewer,
We would like to thank you all for your constructive feedback about the publication.
As you suggested, to improve the integrity of the review we decided to add flow charts in the “2.1. Tablet” section as it is a longer and complex one. Three flow chart was inserted as a figure (Figure 5, 9 and 11) which summarize the main milestones in every year or in a period.

Reviewer 3 Report
The manuscript thoroughly reviewed the fabricated 3D-printed drug delivery systems reported in the literature. There are some comments that may help improve the manuscript to be more attractive to the readers shown as follows:
1. The authors should summarize advantages and limitations of each 3D printed drug delivery system (DDS) compared with DDS prepared by conventional method.
2. Future perspective should be mentioned at the end of the manuscript.
3. Please carefully check typographic errors and incorrect grammars throughout the manuscript.
Author Response
Reply to Reviewer 3
Dear Reviewer,
We would like to thank you all for your constructive comments in the review. Your comments provided valuable insights to refine its contents and analysis. In this document, we tried to address the issues raised as best as possible. Our revision in the manuscript was highlighted by green color.
- The authors should summarize advantages and limitations of each 3D printed drug delivery system (DDS) compared by conventional method.
Thank you for your comment. The advantages and limitations could be found at the end of every drug delivery system.
- Future perspective should be mentioned at the end of the manuscript.
Thank you for your question. A new section was made “3. Future perspective” which hopefully improved the readability and novelty of this work.
- Please carefully check typographic errors and incorrect grammars throughout the manuscript.
Thank you for your comment, as it is a long review we tried to eliminate the spell mistakes as much as possible, but we would like to ask English editing for the thorough check.

Round 2
Reviewer 2 Report
I think it is suitable to be published on Pharmaceutics.
Reviewer 3 Report
The manuscript is now acceptable for publication.